# Women are more likely to expect social sanctions for open defecation: Evidence from Tamil Nadu India

Jinyi Kuang[1], Sania Ashraf[1], Alex Shpenev[1], Maryann Greene Delea[2], Upasak Das[3], Cristina Bicchieri[1] *

1 Center for Social Norms and Behavioral Dynamics, University of Pennsylvania, Philadelphia, PA, United States of America, 2 Gangarosa Department of Environmental Health & Hubert Department of Global Health, Rollins School of Public Health, Emory University, Atlanta, GA, United States of America, 3 Global Development Institute, University of Manchester, Manchester, United Kingdom

* cb36@sas.upenn.edu

**Data Availability Statement:** The dataset analyzed for this study, analysis files and survey items in both English and Tamil can be found in the OSF repository https://osf.io/kysqx/. This study drew

## Abstract

Social sanctions can be effective for sustaining beneficial norms by harnessing the power of social pressure and peer monitoring. Yet, field evidence regarding how norms might be linked to perceived risk of sanction is limited. In this study, we focused on communities located in peri-urban areas of Tamil Nadu, India, and examined how people's perceived prevalence of a socially desirable behavior (i.e., toilet use) relates to the perceived risk of sanctions for deviating from this behavior (i.e., open defecation) in the sanitation domain. Cross-sectional data from 2427 participants in 75 communities revealed that the majority (77%, n = 1861) perceived the risk of informal sanctions related to open defecation. Among those, verbal reprimand was the most common (60%), followed by advice (30%) and gossip (7%). Results from multilevel logistic regression indicated that those who believed toilet use was prevalent in their community were more likely to perceive the risk of social sanctions for open defecation. Moderation analysis revealed that this relationship was robust among women, but attenuated among men. Our findings suggest that women are more likely to expect social sanctions if they deviate from what is perceived as the prevalent sanitation behavior (e.g., toilet use) in their community. Open defecation practices are known to cause psychosocial stress among women due to their experiences with sanitation insecurity, which may include fear of disapproval from community members. Our results highlight the need for gendered intervention strategies when sanitation programs leverage social influence for behavior change.

## 1 Introduction

Informal sanctions, such as gossip and shame, can be powerful in enforcing norms by deterring individuals from engaging in undesirable behaviors [1–8]. Recent findings from laboratory experiments suggest that an individual's propensity to sanction someone who deviates from the norm is related to the prevalence of the desirable behavior [9, 10]. Specifically, when a socially desirable behavior becomes more prevalent among a social group, group members

cross-sectional data (N = 2427) collected between January 11th, 2020 to March 19th, 2020 from an ongoing Cluster Randomized Controlled Trial (CRT). The trial was paused after March 19th, 2020 due to COVID-19 and resumed in June, 2020 to reach a sample size of 2571. The additional 144 subjects did not substantively change the results of the analyses.

**Funding:** This study and its publication are funded by the Bill and Melinda Gates Foundation (Grant No: INV-009118 / OPP1157257). Kantar Republic (https://www.kantar.com/) managed sample selection and data collection. The funders had no role in study design, data collection and analysis, decision to publish, or preparation of the manuscript.

**Competing interests:** The authors have declared that no competing interests exist.

consider the deviators more deserving of punishment [10]. The effect of commonness on the willingness to sanction can be explained by Social Impact Theory [11] and the Social Influence Model [12]. According to these theoretical models, the degree to which social influence affects an individual's beliefs or behaviors increases in proportion to the number of people who act in a uniform way or express uniform agreement on beliefs or behaviors [11, 12]. In this case, as more people behave in what is judged as a socially desirable way, perceived peer pressure on deviators intensifies. Norm internalization provides another perspective that offers an explanation of the association between perceived prevalence of behavior and the likelihood of perceiving the risk of sanctions. According to norm internalization, when a behavior is commonly practiced and approved by members of a social group, the norm may be internalized as 'moral' and norm-abiding behavior will be considered good or appropriate [13, 14]. Consequently, people expect informal external sanctions, such as shaming or gossiping, or experience internal ones, such as guilt or shame, when deviating from the norm [15, 16]. Taking these two perspectives together, we expect to observe a positive relationship between commonness and perceived risk of sanction in a natural setting. Specifically, as individuals perceive more people engaging in socially desirable behaviors, they should also be more likely to perceive the risk of sanctions for deviating from how the majority behaves.

Gender differences have been noted in both perceived risk of sanctions and conformity. Literature on the deterrence effect suggests that women tend to perceive a higher risk of sanctions, including shame and embarrassment, than men [17–20]. Women are also more sensitive to social influence, including the peer pressures associated with deviating from common behaviors of their social groups [21, 22]. It is likely that the commonness of behaviors increases the cost of deviation, and women could be more sensitive to the increased potential social cost. We therefore expect to observe a gender difference in the relationship between perceived behavior prevalence and the perceived risk of sanctions for deviating from such behavior. Specifically, this relationship is likely to be stronger for women than for men.

This study utilizes data from an ongoing sanitation behavior change trial in India to assess whether these relationships are observed in close-knit communities and whether there are differences across genders. Open defecation (OD) in India has drawn global attention for its negative impacts on health, economics, and human rights [23]. Substantial field evidence indicates that OD practices are related to increased psychosocial stress due to decreased privacy, increased risk of sexual harassment, and potential social sanctions such as gossip, particularly among women [24–27]. Using social pressure and peer monitoring can be effective for promoting toilet use [28]. However, such social components could potentially induce psychosocial stress (e.g., shame, guilt, and fear) [29], which might disproportionately affect women. Therefore it is increasingly important to understand how social beliefs about sanitation behaviors relate to social pressure, such as informal sanctions, and to quantitatively assess gender differences in this relationship. Findings from this study contribute empirical evidence from natural, community settings that can be used to refine social norms-focused theories, and help inform the development of gendered interventions [25, 26, 30].

This study aims to examine the correlation between two types of social beliefs: perceived prevalence of sanitation behaviors and perceived risk of OD sanctions in Tamil Nadu India. We hypothesized that 1) the perceived prevalence of toilet use is positively associated with the likelihood of perceiving the risk of sanctions of a transgressing behavior (i.e., OD); and 2) this relationship is moderated by gender. Women are more likely to perceive the risk of sanctions when they deviate from the behaviors practiced by the majority of their community members (i.e., toilet use), whereas this relationship is attenuated among men.

## 2 Methods

### 2.1 Data collection

Trained fieldworkers administered surveys in 2427 households in 75 peri-urban communities in Tamil Nadu, India to collect data from January 11th, 2020 to March 19th, 2020. Peri-urban refers to areas often at the periphery of cities [31]. Households in peri-urban areas of Tamil Nadu often have living structures that are close to each other and maintain close interpersonal relationships [32, 33]. Respondents were randomly selected from a complete listing of dwelling units/households in each sampling unit. We used sampling units (i.e., census wards) as proxies for communities. Further details regarding the sampling strategy can be found in the trial protocol [34].

All survey items were translated from English to the local language, Tamil, and back-translated into English by a third party to ensure its validity. We then piloted the questionnaire among 70 participants from communities similar to our study site to test for comprehension. Next we conducted extensive training sessions in which all enumerators received training from bilingual, experienced trainers to ensure the standard survey collection procedure. Any feedback received or revisions occurred during the training phase were reviewed and approved by a Tamil speaking supervisor to ensure quality. Prior to the survey administration, field workers obtained oral consent and provided a written version to all participants. The data were collected by Computer Assisted Personal Interviewing (CAPI) on hand-held tablets. This study was approved by the University of Pennsylvania Institutional Review Board (Protocol #: 833854) which served as the central IRB. We also received the approval from the Social Research Institute (IRB registration number: IORG0009562) in India, which served as the local IRB at our study site.

### 2.2 Measurement

To measure the perceived risk of social sanctions for OD, we asked respondents, *"If someone from your community defecated in the open, would anyone do or say anything in response to that?"*. Those who responded *yes* were asked, *"What would be done in response to someone who defecated in the open in your community?"*. Based on the previous formative study findings, the response options included various intensities of sanctions, such as advice, verbal reprimand, gossip, fines, loss of public benefits such as ration cards, and violence [26, 33]. To measure the perceived prevalence of toilet use, we asked respondents, *"Think about ten members of your community. Out of them, how many do you think use a toilet every time to defecate?"*. The answer ranged from 0 to 10, where 0 represented not prevalent at all, and 10 represented extremely prevalent [35, 36]. We also collected respondents' demographic characteristics, including age, years of schooling, marital status, caste, primary defecation practices a week prior to the survey, and household economic status.

### 2.3 Analysis

To test whether the perceived prevalence of toilet use was associated with perceived risk of sanctions for OD, we used a multilevel logistic regression with perceived risk of sanctions as the dependent variable (coded as a binary variable, 1 = perceiving the risk of OD sanctions, 0 = not perceiving the risk of OD sanctions), and perceived prevalence of toilet use as the independent variable. We included respondents' primary defecation behavior (coded as a binary variable, 1 = use toilets, 0 = open defecation), gender, age, years of schooling, marital status (coded as a binary variable with 1 = currently married, 0 = single/divorced/widow), caste, and household economic status (proxied by the number of relevant household assets owned,

ranging from 0 to 10) as covariates, and accounted for the mixed effects of communities where respondents resided. To test whether the association between perceived risk of social sanctions for OD and perceived prevalence of toilet use differed between women and men, we conducted a moderation analysis. We used the perceived risk of OD sanctions as the binary dependent variable, the perceived prevalence of toilet use as the independent variable, and gender as a moderator. The analyses were conducted in R version 3.6, and a significance level of 5% was used.

## 3 Results

A total of 2427 respondents (women = 53%) were enrolled in the study. The average age of respondents was 45 years (SD = 16), ranging from 18 to 98. Among surveyed respondents, 98% were Hindu, 81% had access to any type of toilet, and 75% reported using a toilet primarily in the two days before the survey. The demographic characteristics, defecation practices, and social beliefs of the study population are described in Table 1. The majority (77%, n = 1861) expected a social sanction related to OD. Among those, verbal reprimand was the most common form of social sanction (60%), followed by advice (30%), gossip (7%), and violence (3%).

Our results showed that those who perceived toilet use as more prevalent in their communities were more likely to perceive the risk of social sanctions for OD (OR = 1.16, 95% CI: 1.11, 1.22, p < 0.001), which is consistent with hypothesis 1 (Table 2). Those who primarily used a toilet in the two days before the survey were more likely to perceive risk of social sanctions for open defecation (OR = 2.22, 95% CI: 1.66, 2.96, p < 0.001). Compared to men, women were nearly twice as likely to perceive risk of sanctions for OD (OR = 1.77, 95% CI: 1.41, 2.21, p < 0.001).

Moderation analysis showed a gender difference in the relationship between the perceived prevalence of toilet use and the perceived risk of OD sanctions (OR = 1.19, 95% CI: 1.09, 1.29, p < 0.001). This interaction is illustrated in Fig 1. Post-hoc probing showed that women were more likely to perceive the risk of social sanctions for OD when they believed more people

**Table 1. Demographic characteristics, defecation practices, and social beliefs of the study population, Tamil Nadu, India 2020.**

| | Total (N = 2427) | Women (N = 1277) | Men (N = 1150) |
|---|---|---|---|
| **Age mean (sd)** | 44.7 (16.0) | 44.7 (16.0) | 44.7 (16.1) |
| **Years of schooling mean (sd)** | 7.9 (5.2) | 7.0 (5.4) | 9.0 (4.8) |
| **Currently married n (%)** | 1738 (71.6) | 851 (66.6) | 887 (77.1) |
| **Caste n (%)** | | | |
| General | 298 (12.3) | 69 (5.4) | 229 (19.9) |
| Scheduled caste | 474 (19.5) | 319 (25.0) | 155 (13.5) |
| Scheduled tribe | 22 (0.9) | 15 (1.2) | 7 (0.6) |
| Other backward caste | 1633 (67.3) | 874 (68.4) | 759 (66.0) |
| **Religion n (%)** | | | |
| Hindu | 2375 (97.9) | 1249 (97.8) | 1126 (97.9) |
| Muslim | 21 (0.9) | 12 (0.9) | 9 (0.8) |
| Christian | 31 (1.3) | 16 (1.3) | 15 (1.3) |
| **Socio-economic status mean (sd)** | 4.5 (1.8) | 4.4 (1.8) | 4.7 (1.7) |
| **Had access to a toilet n (%)** | 1955 (80.6) | 943 (73.8) | 1012 (88.0) |
| **Reported primarily using toilets in the past 2 days n (%)** | 1819 (74.9) | 884 (69.2) | 935 (81.3) |
| **Reported perceiving the risk of sanctions for open defecation n (%)** | 1861 (76.7) | 1006 (78.8) | 855 (74.3) |
| **Perceived prevalence of toilet use mean (sd)** | 7.4 (2.6) | 7.2 (2.8) | 7.6 (2.2) |

**Table 2. Multivariable analysis of factors associated with perceived risk of social sanction for open defecation, Tamil Nadu, India, 2020.**

| | Perceived risk of sanctions for open defecation | |
|---|---|---|
| | Odds ratio (95% CI) | |
| | **Without interaction** | **With interaction** |
| Perceived prevalence of toilet use | 1.16\*\*\* (1.11, 1.22) | 1.04 (0.97, 1.12) |
| Gender (Ref. Men) | | |
| Women | 1.77\*\*\* (1.41, 2.21) | 0.54\* (0.29, 1.00) |
| Perceived prevalence x Women | | 1.19\*\*\* (1.09, 1.29) |
| Primary defecation behavior (Ref. Open defecation) | | |
| Use toilets | 2.22\*\*\* (1.66, 2.96) | 2.25\*\*\* (1.68, 3.01) |
| Age | 0.99 (0.99, 1.00) | 0.99 (0.96, 1.00) |
| Marital status (Ref. Not currently married) | | |
| Currently married | 0.74\*\* (0.58, 0.95) | 0.73\*\* (0.57, 0.94) |
| Social economic status | 1.04 (0.97, 1.12) | 1.05 (0.98, 1.13) |
| Caste (Ref. General caste) | | |
| Scheduled caste | 0.48\*\*\* (0.29, 0.80) | 0.49\*\*\* (0.29, 0.81) |
| Scheduled tribe | 0.11\*\*\* (0.04, 0.30) | 0.10\*\*\* (0.04, 0.30) |
| Other backward caste | 0.38\*\*\* (0.24, 0.58) | 0.36\*\*\* (0.24, 0.56) |
| Years of schooling | 1.00 (0.97, 1.03) | 1.00 (0.97, 1.03) |

Note: We used multilevel logistic regression with community cluster mixed effect. The total number of observations is 2427. Stars denote adjusted p-values.

\*p<0.05

\*\*p<0.01

\*\*\*p<0.001.

used a toilet (OR = 1.24, 95% CI: 1.17, 1.32, p < 0.001). However, this association was lower among men, and was not statistically significant (OR = 1.04, 95% CI: 0.97,1.12, p = 0.24). This gender difference is robust while controlling for toilet use in the two days before the survey, women's social status (proxied by age and marital status), years of education, caste, household economic status, and community of residence.

## 4 Discussion

This study examines the relationship between the perceived prevalence of socially desirable sanitation behaviors and the perceived risk of sanctions for deviation (i.e., OD), as well as the gender difference in this relationship in the context of defecation behaviors in India. We found those who perceived toilet use as more prevalent were more likely to perceive the risk of sanctions for deviating from what the majority does (i.e., OD). Our finding is consistent with previous laboratory experiments in which people perceived the deviator as more deserving of punishment when the socially desirable behaviors were more common [10]. We also found that women were more likely to expect social sanctions for OD when they perceived more community members using a toilet. One perspective that offers an explanation is that women tend to be more relationship-oriented than men [37, 38], especially in close-knit communities in which behaving in a deviant way could hamper the development of social ties or the mainte-nance of interpersonal relationships with others.

Our findings contribute to the growing evidence about women's vulnerability to the psy-chosocial stress related to sanitation and their experiences with sanitation insecurity [25, 26, 30]. Findings from other studies indicate that OD among women may experience fear and

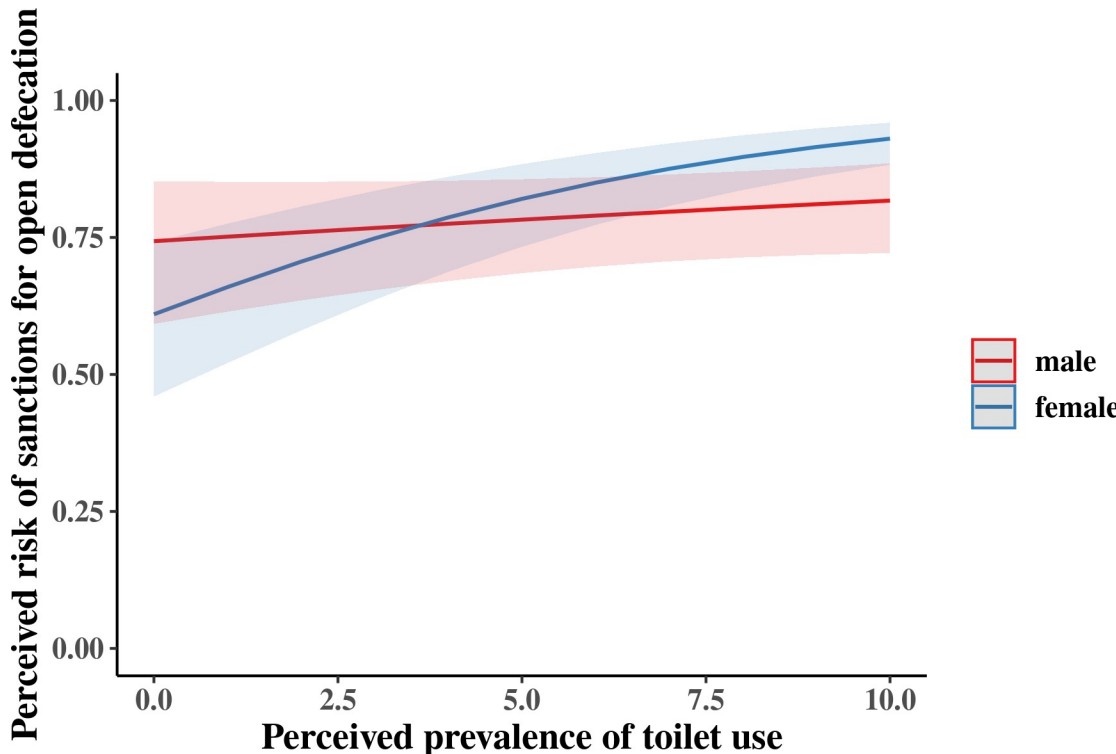

**Fig 1. The average marginal effect of perceived prevalence of toilet use on the perceived risk of sanctions for open defecation by gender, Tamil Nadu, India, 2020.**

worry of disapproval from their community members. OD spots are commonly at a distance from people's residences and often provide little privacy for women. In India, it is often considered a dishonor to one's family if women are seen practicing OD [27]. Having to defecate in the open could indicate absence or inadequate access to sanitation, which has overarching impacts on women's sanitation and menstrual hygiene management across life stages [30]. Inadequate menstrual hygiene management is also tied to school absenteeism for girls, which highlights the disproportionate impact on women who do not have adequate access to sanitation [39]. These negative experiences, both external, such as shaming or gossip, as well as internal, such as guilt or shame, may have long-term impacts on mental health and well-being [40]. Therefore, understanding gender differences in the social beliefs related to sanitation behaviors and how these social beliefs relate to each other can inform gendered sanitation interventions. Program implementers and policymakers that aim to promote beneficial behaviors, such as toilet use in close-knit communities, should consider potential psychosocial stressors and take gendered approaches.

In addition, we assessed the dynamics of people's social beliefs in communities by extending the application of theories beyond the traditional laboratory experiment setting. The empirical evidence collected from study participants in their natural community settings as generated by this study contributes to the advancement and application of theories related to social norms.

Our study has a few limitations. First, the cross-sectional study design only allows for examinations of the correlation between the social beliefs of perceived prevalence of sanitation behaviors and perceived risk of OD sanctions. Future research with a longitudinal design is needed to better understand the complex relationships among different social beliefs and the

implications of gender differences, accounting for contextual factors. Qualitative research can also provide context and insight into how these beliefs manifest in the local context to inform specific behavior change content or strategies. Second, respondents might underreport their OD practice due to social desirability bias, given the substantial national and regional programs that encourage toilet use in our study areas. Those who reported perceiving risks of social sanctions for OD could also be more likely to underreport their OD practice to avoid negative feelings. Therefore the positive correlation between toilet use and perceived risk of sanctions could be smaller than estimated. Third, our survey measures were developed and validated in similar peri-urban communities in Tamil Nadu, India, which could be context-specific. However, evidence from rural Odisha (India) that highlighted similar psychosocial stressors suggests these measurements and findings may be applicable in similar contexts [40].

To conclude, our findings suggest that women are more likely to expect social sanctions for deviating from what is perceived as the prevalent sanitation behavior (e.g., toilet use) in their communities. Open defecation practices may bring about psychosocial stress among women due to increased sanitation-related insecurity, including the fear of disapproval from their community members. While leveraging social influence and peer monitoring can be effective in promoting toilet use, behavior change strategies that include such intervention components should be gendered and context-specific.

## Acknowledgments

We thank Sarah Girard for her feedback on the manuscript. We acknowledge our data collection partner, Kantar Public in Delhi, India, and the field workers who conducted data collection for this study. We are also grateful for the time of the respondents who voluntarily contributed to answering the survey questions.

## Author Contributions

**Conceptualization:** Jinyi Kuang, Alex Shpenev.

**Data curation:** Jinyi Kuang, Sania Ashraf, Alex Shpenev, Maryann Greene Delea, Upasak Das, Cristina Bicchieri.

**Formal analysis:** Jinyi Kuang.

**Funding acquisition:** Cristina Bicchieri.

**Investigation:** Jinyi Kuang, Sania Ashraf, Alex Shpenev, Maryann Greene Delea, Upasak Das.

**Methodology:** Jinyi Kuang, Alex Shpenev.

**Project administration:** Jinyi Kuang, Sania Ashraf, Alex Shpenev, Maryann Greene Delea, Upasak Das.

**Resources:** Cristina Bicchieri.

**Software:** Jinyi Kuang.

**Supervision:** Sania Ashraf, Maryann Greene Delea, Upasak Das, Cristina Bicchieri.

**Validation:** Sania Ashraf, Maryann Greene Delea.

**Visualization:** Jinyi Kuang.

**Writing – original draft:** Jinyi Kuang.

**Writing – review & editing:** Jinyi Kuang, Sania Ashraf, Alex Shpenev, Maryann Greene Delea, Upasak Das, Cristina Bicchieri.

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
