## [Decision Letter · Decision Letter 0]

4 Sep 2020

PONE-D-20-24216

Women are more sensitive to perceiving the risk of social sanctions for deviating from the norm: field evidence from sanitation practices in Tamil Nadu, India

PLOS ONE

Dear Dr. Kuang,

Thank you for submitting your manuscript to PLOS ONE. After careful consideration, we feel that it has merit but does not fully meet PLOS ONE’s publication criteria as it currently stands. Therefore, we invite you to submit a revised version of the manuscript that addresses the points raised during the review process.

Please find below the reviewer's comments, as well as those of mine.

We look forward to receiving your revised manuscript.

Kind regards,

Valerio Capraro

Academic Editor

PLOS ONE

Additional Editor Comments:

I have now collected one review from one expert in the field. The review is very positive and suggests a minor revision. I have read the paper and I agree with the reviewer. Therefore, I would like to invite you to revise your article for Plos One. Also, please include the instructions in the Appendix.

I am looking forward for the final version.

Journal Requirements:

2. During our internal checks, the in-house editorial staff noted that you conducted research or obtained samples in another country. Please check the relevant national regulations and laws applying to foreign researchers and state whether you obtained the required permits and approvals from India. Please address this in your ethics statement in both the manuscript and submission information. In addition, please ensure that you have suitably acknowledged the contributions of any local collaborators involved in this work in your authorship list and/or Acknowledgements. Authorship criteria is based on the International Committee of Medical Journal Editors (ICMJE) Uniform Requirements for Manuscripts Submitted to Biomedical Journals - for further information please see here: https://journals.plos.org/plosone/s/authorship.

4. In the Methods, please discuss whether and how the questionnaire was validated and/or pre-tested after translation. If this did not occur, please provide the rationale for not doing so.

5. The reference to caste/ backward caste may be demeaning and derogatory. Please note that according to our submission guidelines (http://journals.plos.org/plosone/s/submission-guidelines), potentially stigmatizing labels should be deleted or changed to more current, acceptable terminology.

Reviewers' comments:

Reviewer's Responses to Questions

**Comments to the Author**

1. Is the manuscript technically sound, and do the data support the conclusions?

Reviewer #1: Partly

2. Has the statistical analysis been performed appropriately and rigorously? 

Reviewer #1: Yes

3. Have the authors made all data underlying the findings in their manuscript fully available?

Reviewer #1: No

4. Is the manuscript presented in an intelligible fashion and written in standard English?

Reviewer #1: Yes

5. Review Comments to the Author

Reviewer #1: Summary

The paper presents results from a survey on open defecation in a poor area of India. The authors study correlation between perceived prevalence of toilet use (strength of the social norm of using toilets instead of open defecation) and perceived risk of sanctions for open defecation. The authors find that people who perceive the norm of toilet use to be stronger, also perceive the risk of social sanctions when not committing to the norm to be higher. The result is driven by women’s perceptions and not by men’s.

Comments

1. The paper is well written. However, the authors should stress more that their main result is of a purely correlational nature. The authors mention it in their discussion, but pointing out the limitations earlier would be useful too.

2. The title is somewhat confusing. The authors could try to make it more understandable – without reading the paper it is not clear what the authors mean.

3. I would recommend to cut the last two sentences in the abstract.

4. “Those who primarily used a toilet in the two days before the survey were more likely to perceive risk of social sanctions for open defecation (OR=2.22, 95% CI:1.66, 2.96, p<0.001).” – some participants who perceive a higher risk of social sanctions for open defecation might be lying when answering whether they used a toilet in the last days. The authors should acknowledge the possibility of lying in the text.

6. PLOS authors have the option to publish the peer review history of their article (what does this mean?). If published, this will include your full peer review and any attached files.

Reviewer #1: No

---

## [Author Response · Author response to Decision Letter 0]

24 Sep 2020

Valerio Capraro

Academic Editor 

PLOS ONE

September 15, 2020

Dear Dr. Capraro,

Thank you for reviewing our manuscript entitled "Women are more sensitive to perceiving the risk of social sanctions for deviating from the norm: field evidence from sanitation practices in Tamil Nadu, India" (manuscript #: PONE-D-20-24216). Your comments and those of the reviewer were highly insightful and enabled us to greatly improve the quality of our manuscript. We appreciate the suggested modifications and have revised the manuscript accordingly. 

In the remainder of this document, yours and the reviewer’s comments are shown in bold, our response is shown in plain typeface, and sections from the revision are contained within quotation marks.

We also made minor changes in the results sections to improve clarity of the reporting. Specifically, we 1) fixed the rounding errors and typos; 2) replace the post-hoc subgroup analysis by women and men with post-hoc probing with women or men as baseline groups. Those changes did not significantly alter the results of our analysis but provide clarity. We hope that the revised manuscript and our accompanying responses will be sufficient to make our manuscript suitable for publication in PLOS ONE.

Thanks very much for your support, and we look forward to hearing from you soon.

Sincerely,

Jinyi Kuang

Editor:

1. Please ensure that your manuscript meets PLOS ONE's style requirements, including those for file naming. The PLOS ONE style templates can be found at: https://journals.plos.org/plosone/s/file?id=wjVg/PLOSOne_formatting_sample_main_body.pdf and

Revised.

2. During our internal checks, the in-house editorial staff noted that you conducted research or obtained samples in another country. Please check the relevant national regulations and laws applying to foreign researchers and state whether you obtained the required permits and approvals from India. Please address this in your ethics statement in both the manuscript and submission information. In addition, please ensure that you have suitably acknowledged the contributions of any local collaborators involved in this work in your authorship list and/or Acknowledgements. Authorship criteria is based on the International Committee of Medical Journal Editors (ICMJE) Uniform Requirements for Manuscripts Submitted to Biomedical Journals - for further information please see here: https://journals.plos.org/plosone/s/authorship.

Thanks for raising this concern. We obtained approval from the local IRB in India prior to data collection. We added the following text to the ethical statement in both the manuscript and submission information:

“This study was approved by the University of Pennsylvania Institutional Review Board (Protocol #: 833854) which served as the central IRB. We also received the approval from the Social Research Institute (IRB registration number: IORG0009562) in India, which served as the local IRB at our study site.”

We acknowledged our data collection partner, Kantar Public in Delhi, India, and the field workers who conducted data collection for this study in the acknowledgment section. 

3. Please include additional information regarding the survey or questionnaire used in the study and ensure that you have provided sufficient details that others could replicate the analyses. For instance, if you developed a questionnaire as part of this study and it is not under copyright more restrictive than CC-BY, please include a copy, in both the original language and English, as Supporting Information.

We provided the survey questions and choice options in English in the measurement section. To provide further details and ensure the replicability of our research, we uploaded a copy of the survey items used in this study in both English and the local language (Tamil) as supporting materials to the OSF repository along with the dataset and analysis files. The link can be found in the data availability section. As reported in the method section, this study draws the cross-sectional data from an ongoing Cluster Randomized Trial (CRT). We will make all datasets and instruments of the overarching CRT available to the public in their entirety after the completion of the CRT.

4. In the Methods, please discuss whether and how the questionnaire was validated and/or pre-tested after translation. If this did not occur, please provide the rationale for not doing so.

 To clarify this point we added the following:

 “All survey items were translated from English to the local language, Tamil, and back-translated into English by a third party to ensure its validity. We then piloted the questionnaire among 70 participants from communities similar to our study site and conducted tests for comprehension. Next, we conducted extensive training sessions in which all enumerators received training from bilingual, experienced trainers to ensure a standard survey collection procedure. Any feedback and revisions during the training phase were reviewed and approved by a bilingual supervisor to ensure its validity.”

5. The reference to caste/ backward caste may be demeaning and derogatory. Please note that according to our submission guidelines (http://journals.plos.org/plosone/s/submission-guidelines), potentially stigmatizing labels should be deleted or changed to more current, acceptable terminology.

Thanks for pointing this out. We share your concern and we understand that these terminologies may sound demeaning. However, these are the official terminologies currently used by the Government of India [1], Government of Tamil Nadu [2], other national-level surveys including the National Family Health Survey (NFHS) [3,4], and peer-reviewed studies from India [5,6]. We adopted these terminologies in our survey to ensure it fit the local context, is understood by the respondents and enabled our results to be replicable and comparable with other studies in similar contexts. 

Reviewer #1 Comments:

1. The paper is well written. However, the authors should stress more that their main result is of a purely correlational nature. The authors mention it in their discussion, but pointing out the limitations earlier would be useful too.

Thanks for the suggestions. We used “association” throughout the manuscript to emphasize the correlational nature of the relationship between the perceived risk of sanctions for OD and the perceived prevalence of toilet use. In addition to specifying this as a limitation, we added the “correlational” term in our study aim to further clarify the correlational nature of these two types of social beliefs.

“This study aims to examine the correlation between two types of social beliefs: perceived prevalence of sanitation behaviors and perceived risk of OD sanctions in Tamil Nadu India.”

2. The title is somewhat confusing. The authors could try to make it more understandable – without reading the paper it is not clear what the authors mean.

We revised the title to “Women are more likely to expect social sanctions for open defecation: evidence from Tamil Nadu India”

3. I would recommend to cut the last two sentences in the abstract.

Thanks for the suggestion. We think the last two sentences in the abstract reflect some key discussion points and the implications of this study. It linked to prior studies that highlighted women’s vulnerability to sanitation related psychosocial stress. Given the wide readership of PLOS ONE, we believe these discussion points would be of particular interest to program implementers and policymakers who aim to promote toilet use with a social norm focused approach. 

4. “Those who primarily used a toilet in the two days before the survey were more likely to perceive risk of social sanctions for open defecation (OR=2.22, 95% CI:1.66, 2.96, p<0.001).” – some participants who perceive a higher risk of social sanctions for open defecation might be lying when answering whether they used a toilet in the last days. The authors should acknowledge the possibility of lying in the text.

Thanks for pointing this out. We add the following text to the limitation section:

“Second, respondents might underreport their OD practice due to social desirability bias, given the substantial national and regional programs that encourage toilet use in our study areas. Those who reported perceiving risks of social sanctions for OD could also be more likely to underreport their OD practice to avoid negative feelings. Therefore the positive correlation between toilet use and perceived risk of sanction could be smaller than estimated.”

References 

1. National Commission for Backward Classes. [cited 5 Sep 2020]. Available: http://www.ncbc.nic.in/Home.aspx?ReturnUrl=%2f

2. Government of Tamil Nadu. List of backward classes approved by the Government of Tamil Nadu. [cited 5 Sep 2020]. Available: http://www.bcmbcmw.tn.gov.in/bclist.htm

3. Vart P, Jaglan A, Shafique K. Caste-based social inequalities and childhood anemia in India: results from the National Family Health Survey (NFHS) 2005–2006. BMC Public Health. 2015;15: 537. doi:10.1186/s12889-015-1881-4

4. India - National Family Health Survey 2015-2016. [cited 5 Sep 2020]. Available: https://microdata.worldbank.org/index.php/catalog/2949/datafile/F29/V1573

5. Shaikh M, Miraldo M, Renner A-T. Waiting time at health facilities and social class: Evidence from the Indian caste system. PLOS ONE. 2018;13: e0205641. doi:10.1371/journal.pone.0205641

6. Subramanyam MA, Kawachi I, Berkman LF, Subramanian SV. Socioeconomic Inequalities in Childhood Undernutrition in India: Analyzing Trends between 1992 and 2005. PLOS ONE. 2010;5: e11392. doi:10.1371/journal.pone.0011392

---

## [Editor Report · Decision Letter 1]

28 Sep 2020

Women are more likely to expect social sanctions for open defecation: evidence from Tamil Nadu India

PONE-D-20-24216R1

Dear Dr. Kuang,

We’re pleased to inform you that your manuscript has been judged scientifically suitable for publication and will be formally accepted for publication once it meets all outstanding technical requirements.

Kind regards,

Valerio Capraro

Academic Editor

PLOS ONE
---

## [Editor Report · Acceptance letter]

5 Oct 2020

PONE-D-20-24216R1 

Women are more likely to expect social sanctions for open defecation: evidence from Tamil Nadu India 

Dear Dr. Kuang:

I'm pleased to inform you that your manuscript has been deemed suitable for publication in PLOS ONE. Congratulations! Your manuscript is now with our production department. 

Kind regards, 

on behalf of

Dr. Valerio Capraro 

Academic Editor

PLOS ONE